# Development of Genistein Drug Delivery Systems Based on Bacterial Nanocellulose for Potential Colorectal Cancer Chemoprevention: Effect of Nanocellulose Surface Modification on Genistein Adsorption

**DOI:** 10.3390/molecules27217201

**Published:** 2022-10-24

**Authors:** Melissa Castaño, Estefanía Martínez, Marlon Osorio, Cristina Castro

**Affiliations:** 1School of Engineering, Universidad Pontificia Bolivariana, Circular 1#70-01, Medellín 050031, Colombia; 2School of Health Science, Biology Systems Research Group, Universidad Pontificia Bolivariana, Calle 78b #72a-159, Medellín 050031, Colombia

**Keywords:** bacterial nanocellulose, surface modification, genistein, controlled drug delivery system, colorectal cancer

## Abstract

Genistein is an isoflavone with antioxidant, anti-inflammatory, and anticancer properties. That said, its use in the industry is limited by its low solubility in aqueous systems. In this work, bacterial nanocellulose (BNC) and BNC modified with cetyltrimethylammonium (BNC-CTAB) were evaluated as genistein-encapsulating materials for their controlled release in cancer chemoprevention. Thin films were obtained and characterized by contact angle, AFM, TEM, UV–Vis spectroscopy FTIR, and TGA techniques to verify surface modification and genistein encapsulation. The results show a decrease in hydrophilization degree and an increase in diameter after BNC modification. Furthermore, the affinity of genistein with the encapsulating materials was determined in the context of monolayer and multilayer isotherms, thermodynamic parameters and adsorption kinetics. Spontaneous, endothermic and reversible adsorption processes were found for BNC-GEN and BNC-CTAB-GEN. After two hours, the maximum adsorption capacity corresponded to 4.59 mg GEN∙g^−1^ BNC and 6.10 mg GEN∙g^−1^ BNC-CTAB; the latter was a more stable system. Additionally, in vitro release assays performed with simulated gastrointestinal fluids indicated controlled and continuous desorption in gastric and colon fluids, with a release of around 5% and 85%, respectively, for either system. Finally, the IC50 tests made it possible to determine the amounts of films required to achieve therapeutic concentrations for SW480 and SW620 cell lines.

## 1. Introduction

Annually, there are 9.0 million cancer deaths worldwide [1], and colorectal cancer (CRC) ranks third in the world and fourth with new cases reported [2,3]. The treatments for CRC include endoscopic resection, polypectomy, a partial colectomy, chemotherapy, and radiation therapy. All of these are invasive and are related to side effects [4,5]. Consequently, researchers seek to develop new strategies that can increase the specificity of treatments, reduce side effects, and prevent the formation and progression of cancer [6,7]. In recent years, novel treatments have focused on natural compounds as an alternative for cancer prevention and treatments due to their nontoxic and selective character [8].

Genistein, or 4′,5,7-trihydroxyisoflavone, is a polyphenolic compound member of the isoflavones family that is derived from different food sources such as soybeans, legumes, broad beans, and lupins [9]. Soy isoflavones are known as phytoestrogens, due to the structural similarity between them and 17β-estradiol; in the case of genistein, carbons 4 and 7 on the phenol rings are similar to the OH groups of estradiol and form bonds with estrogen receptor (ER) residues. Carbon 7 binds to His475, and carbon 4 binds with Arg346 and Glu305; these connections allow GEN to bind with isoforms α and β of ER and stimulate estrogen [10,11], making genistein a chemopreventive agent against cancer, especially in breast, prostate, and CRC [10]. It also influences inflammation, cell proliferation and the modulation of epigenetic changes, and prevents angiogenesis and has antioxidant activity toward cancer cells [9,10,12]. However, the pharmaceutical use of genistein is limited by its low solubility [13]. Encapsulation technology may provide a means of increasing the biodisponibility of compounds, protecting the bioactive compounds from oxidation, and enhancing their miscibility and absorption in aqueous systems [14].

Cellulose is the most abundant polymer on Earth; it is present in algae, wood, cotton, bacteria, and fungi [15]. Bacterial nanocellulose (BNC) is obtained from different bacteria genera such as *Acetobacter*, *Sarcina*, and *Komagateibacter* [16]. BNC is composed of β-1,4-D(+) glucose units, and it is lignin- and hemicellulose-free, which is an advantage for avoiding chemical isolation treatments [15,16]. Besides this, BNC is made up of a three-dimensional network of nanoribbons with diameters at the nanoscale [15]. Its structure is assembled from inter- and intramolecular hydrogen bonds between nanoribbons, allowing for the formation of structures with a large surface area, open porosity, and good tensile strength [17,18]. Likewise, BNC is biocompatible and nontoxic, and has a high potential in medical applications, such as in the treatment of wounds, the development of artificial skin, tissue regeneration, the manufacture of dental and artery implants, as well as in protein immobilization and controlled drug release [16,19,20,21]. In drug delivery systems, BNC has been used to encapsulate different compounds. For instance, Subtaweesin et al. (2018) loaded curcumin on BNC, and the results showed anticancer activity against A375 melanoma cancer cells and no significant cytotoxic activity against human keratinocytes and human dermal fibroblast [22]. However, some authors have modified the hydrophilic nature of BNC to improve the adsorption of nonpolar bioactive compounds [23]. S. Akagi et al. (2021) prepared cellulose in a culture supplemented with carboxymethylcellulose and hydroxypropylcellulose to carry hydrophobic cancer drugs such as paclitaxel (PTX) formulations, which reduced the side effects of PTX and toxicity and increased the therapeutic efficacy of hydrophobic compounds [24]. M. L. Cacicedo et al. (2016) modified BNC by an in-situ method with alginate addition to load doxorubicin (DOX); the results showed decreased HT-29 viability compared to free DOX and stability in time (14 days) [21]. Finally, Wang et al. (2019) coated plant-derived cellulose nanocrystals with cetyltrimethylammonium bromide (CTAB) to improve the dissolution rate and antioxidant activity of genistein in aqueous systems, with results showing a decrease in genistein crystalline structure, and an increase in the dissolution rate to 72–92% compared with the original genistein, as well as enhanced antioxidant activity [12]. Additionally, Qu et al. (2019) investigated CTAB modification to increase the hydrophobicity and enhance the dispersibility of fibrillated BNC for further applications. The results show that CTAB modification improved the thermal stability and hydrophobicity of fibrillated cellulose [25]. Nevertheless, the literature lacks information on the effect of this surface modification on genistein loaded on BNC, and its application for oral drug delivery in cancer chemoprevention. Therefore, this paper aims to develop drug delivery systems (DDS) based on BNC and BNC-CTAB for genistein, along with the mathematical model of the systems, material characterization, and in vitro studies.

## 2. Results

### 2.1. Calculation of Inhibitory Concentration (IC50) for Free Genistein 

The half maximal inhibitory concentration (IC50) is a measure of the potency or drug efficacy; this indicates the amount of a bioactive compound that is necessary to inhibit a biological process by half. Genistein’s ability to inhibit cancer cell growth was evaluated. These values are key to any subsequent DDS design. Figure 1 shows the results of the inhibitory effects of genistein on SW480, SW620, and HaCaT cell lines. The results show that cell inhibition is time- and dose-dependent, suggesting increased inhibition in all cell lines at 48 h. Previous results in the literature indicate IC50 values of 39.43, 50 and 52.62 µM against CRC cells such as HCT-116, SW620, HT29 and SW480 cells, respectively [26,27,28,29]. Similarly, IC50 values of 127.60 and 140.30 µM demonstrate genistein’s anticancer activities against MCF-7 and Hep3B cells, corresponding to hepatocellular and breast carcinomas [30,31]. Figure 1d summarizes the IC50 values of SW40, SW620, and HaCaT at 24 and 48 h in this study. Our results indicate that genistein has selectivity and anticancer activity against CRC cell lines. In this work, the IC50 values for free genistein were used for the development of oral DDS for the chemoprevention of CRC.

### 2.2. Development of BNC and BNC-CTAB Materials

To improve the adsorption of nonpolar compounds such as genistein, the surface modification of BNC has emerged as a possibility to enhance the solubility and biodisponibilty in aqueous systems. Figure 2 shows the TEM micrographs of BNC before and after CTAB modification. A 3D network with an entangled structure of randomly oriented nanoribbons can be observed. For BNC nanoribbons (Figure 2a,b), the diameters ranged between c.a. 10 and 60 nm (Figure 2e). For BNC-CTAB nanoribbons (Figure 2c,d), these values were between 12 and 85 nm. This increase in diameter is not statistically significant. However, the BNC and BNC-CTAB diameter medians were 28.74 ± 14.72 and 34.47 ± 18.68 nm, respectively. 

The addition of cetyl trimethyl ammonium to BNC implies a hydrophobization of nanoribbons [32]. The contact angles formed between deionized water and dry films of BNC and BNC-CTAB correspond to 33.92° and 60.66°, respectively (Figure 3); the increase was statistically significant and corresponded to c.a. two-fold. The contact angle measurements indicate a decrease in hydrophilicity on the BNC-CTAB surface [33,34].

The FTIR spectra of BNC, BNC-CTAB, and CTAB are shown in Figure 4. For BNC, bands at 3330 and 2890 cm^−1^ are characteristic of O-H stretching and C-H stretching [12,21]. The band at 1640 cm^−1^ is related to residual water, and the peak at 1370 cm^−1^ represents C-H bending [12,35]. In addition, bands at 1430, 1056, and 898 cm^−1^ represent the CH_2_ symmetrical bending, C-O vibration, and C-O-C stretching of the sugar ring, respectively [35,36]. For CTAB, characteristic bands were observed between 2925 and 2850 cm^−1^, related to the symmetric and asymmetric stretching of the C-H bond of the alkyl chain [32]. The BNC-CTAB spectra show new peaks at 2915, 2840 and 1599 cm^−1^; the first two are related to the alkyl chain of CTAB, and the last peak is related to carboxyl groups presented due to (2,2,6,6-Tetramethylpiperidin-1-yl)oxyl (TEMPO) pretreatment. 

Figure 5 shows the results of the thermal analysis of BNC, BNC-CTAB ribbons, and free CTAB. The samples presented a thermal degradation temperature of 265 °C for BNC and 210 °C for BNC-CTAB and CTAB. Surface modification decreased the thermal degradation temperature of cellulose. Moreover, the BNC-CTAB ribbons contained less water than BNC, which is related to the larger amount of residue at higher temperatures in the BNC-CTAB sample. The DTG decomposition curves of BNC and CTAB (Figure 5b) demonstrate the appearance of one peak at 345 °C for BNC and 274 °C for CTAB, similar to what was reported in the literature [32,37]. In contrast, the DTG curve of BNC-CTAB shows two peaks; the first at 254 °C is related to the early degradation of alkyl chains and carboxylic groups, while the latter is found in samples due to TEMPO pretreatment, and the second peak at 337 °C is related to the unmodified parts of cellulose [32]. Finally, the previous results indicate the successful modification of BNC with CTAB.

### 2.3. Adsorption Studies

#### 2.3.1. Adsorption Isotherms

Figure 6 shows the experimental data and their adjustment to monolayer models at 0, 23, and 40 °C, and Table 1 presents the adjustment and parameters of the Langmuir, Freundlich, Sips, and Toth models. The experimental data did not fit to multilayer models and can be seen in Appendix A. The BNC experimental data showed an increase in adsorption capacity with an increase in the genistein concentration, and subsequently, a plateau was reached when the concentration in equilibrium (*C_e_*) was 17.91, 13.90, and 15.45 mg∙L^−1^ for BNC at 0, 23, and 40 °C, respectively. Additionally, the experimental data show the adjustment (R^2^ > 0.95) to the models of Langmuir, Freundlich, Sips, and Toth at the three temperatures; hence, these models adequately describe the adsorption process and indicate monolayer adsorption. Otherwise, the BNC-CTAB experimental data show an increase in adsorption capacity with an increase in the genistein concentration, followed by a plateau when the *C_e_* was 10.68 and 10.81 mg∙L^−1^ at 0 and 23 °C; at 40 °C, unstable adsorption was presented, while an increase in genistein adsorption was followed by a decrease in adsorption capacity, and standard deviation growth was presented. Therefore, it was found that the Langmuir, Freundlich, Sips, and Toth models are appropriately adjusted to temperatures of 0 and 23 °C (R^2^ > 0.95). 

The adsorption isotherms of both systems indicate that the adsorption capacity of adsorbents does not depend on the initial concentration, but rather the available sites for the adsorption of genistein [38]. Likewise, the increment in the Langmuir constant (*K_L_*) indicates the relation between the adsorption capacity and temperature. Moreover, the Freundlich constant provides information on the heterogeneity and irreversibility of the system—when $1/n_{f}$ < 0.1 approaches zero, it implies an increase in the surface heterogeneity, while if $1/n_{f}$ < 0.1 the process is irreversible, and $1/n_{f}$ > 1 indicates an unfavorable adsorption isotherm [38]. Regarding reversibility, the results show that the adsorption of genistein in BNC at 23 °C and BNC-CTAB at 0 and 23 °C is reversible, while at 40 °C, it is irreversible for both systems. This means that later difficulties could arise for genistein release at 40 °C; in addition, the reversibility of the system depends on the temperature. For heterogeneity, an increase in temperature causes changes in the surface of the adsorbents, and as a consequence, the heterogeneity increases. The Freundlich isotherm showed that at 23 °C, BNC-GEN has a more heterogeneous surface than BNC-CTAB-GEN, while for the other temperatures, the results are not comparable. Furthermore, the Sips models indicate that at 23 °C, BNC-GEN shows a heterogenous surface, and BNC-CTAB-GEN a homogeneous surface. The Toth isotherm shows, similarly to the Freundlich, that heterogeneity increases with increasing temperature. At 0 °C, both systems are homogeneous, while at 23 °C, BNC-GEN is more heterogenous.

#### 2.3.2. Determination of Thermodynamic Parameters

The thermodynamic results (Table 2) were obtained from $K_{L}$ and the Van’t Hoff equation (Equation (7)). For BNC-GEN, it was found that as the temperature increases, the spontaneity of the adsorption process increases; the positive value of Δ*H* indicates an endothermic process; likewise, a Δ*H* < 40 kJ∙mol^−1^ indicates physisorption, where the related molecular interactions correspond to hydrogen bonds [39]. Similarly, the parameters for BNC-CTAB-GEN indicate spontaneous adsorption processes, and endothermic processes for BNC-CTAB (40 kJ∙mol^−1^ < Δ*H* < 80 kJ∙mol^−1^), meaning that the adsorption of genistein on BNC is due to a complex reaction, where physisorption predominates, as described by Lyubchik et al. (2020) and Scheufele et al. (2016) for liquid–solid systems [40,41]. In this case, molecular interactions correspond to electrostatic and hydrophobic interactions [41]. For both systems, the Δ*S* value shows an increase in the affinity of genistein molecules for BNC-CTAB and an increase in randomness in the liquid–solid interface of BNC-CTAB-GEN compared to BNC-GEN [41,42].

According to the isotherm models, a greater adsorption capacity (*Q_m_*) for BNC-GEN was reached at 23 °C, while the thermodynamic parameters indicate spontaneous and endothermic adsorption processes at this temperature. For BNC-CTAB-GEN, a greater adsorption capacity was reached at 0 °C; however, the affinity constant decreased at this temperature. Moreover, the thermodynamic parameters indicate spontaneous and endothermic adsorption processes. Hence, for subsequent assays, the selected temperature was 23 °C.

#### 2.3.3. Adsorption Kinetics

For the adsorption kinetics, the temperature was 23 °C, with a genistein concentration of 42.28 mg∙L^−1^, to ensure the formation of the monolayer. Figure 7 and Table 3 show the experimental data and the model fit. For BNC-GEN, the results indicate an increase in adsorption capacity in the first 20 min and a desorption event at 60 min. After 100 min, the results showed the equilibrium. The models showed a low adjustment to the experimental data with R^2^ < 0.800; therefore, these models are not suitable for explaining the adsorption of genistein on BNC over time. The low stability of the compound could be related to the hydrophilic nature of BNC and the formation of hydrogen bonds between water and genistein.

Furthermore, for BNC-CTAB-GEN, rapid adsorption was observed in the first 5 min followed by a desorption event at 10 min, and finally, equilibrium was reached at 20 min. The experimental data show a good adjustment (R^2^ > 0.93) to the pseudo-first- (PFO) and pseudo-second-order (PSO) models, with a larger R^2^ being shown by the PSO model. The CTAB surface modification gives genistein greater stability compared to BNC without modification.

Differences between both kinetics are related to the adsorption mechanism. The BNC-GEN adsorption mechanism is characterized by physisorption, in which adsorption and desorption events take place, while BNC-CTAB-GEN is characterized by a complex physical reaction, which helps the modified system to reach equilibrium faster. Finally, the good adjustment to the pseudo-second-order kinetics model indicates, similar to the Langmuir constants, that the adsorption process is not related to adsorbate concentration, but rather to the available sites on the adsorbent (BNC) [43].

### 2.4. Development of Thin Film Drug Delivery System and Characterization

To analyze the textures of the films, the BNC and BNC-CTAB samples were subjected to AFM. The parameters calculated for the surface roughness analysis of BNC and BNC-CTAB were arithmetical mean height or Sa, and root means square height or Sq (Figure 8). The results show an Sa of 64.25 ± 5.26 and 29.20 ± 3.81 nm, and an Sq of 81.69 ± 4.66 and 36.59 ± 2.46 nm, for BNC and BNC-CTAB, respectively. Therefore, BNC is statically rougher than BNC-CTAB. The incorporation of hydrophobic groups into modified cellulose causes the fibers to be homogeneously distributed on the surface after oven drying, avoiding nanoribbon agglomeration. Therefore, BNC is rougher than BNC-CTAB, and the difference is statistically significant.

To verify the adsorption of genistein in BNC and BNC-CTAB, Figure 9 shows the spectra of free genistein (GEN), BNC, BNC-GEN, and BNC-CTAB-GEN samples. For genistein, characteristic bands are observed at 3404, 1652, 1517, 1308, 1273, and 840–810 cm^−1^. Due to the low proportion of genistein to the amount of cellulose in the films, the FTIR spectra for BNC-GEN do not show differences from the BNC spectra, which is attributed to the masking of genistein bands by the bands of BNC. However, in BNC-CTAB-GEN, there are changes in the heights and ratios of the bands, mainly at 1630 cm^−1^, and between 1070 and 1040 cm^−1^.

Figure 10 shows the thermal analysis of BNC, GEN, BNC-GEN, and BNC-CTAB-GEN. The samples showed a thermal degradation temperature of 265 °C for GEN and BNC-GEN, and 200 °C for BNC-CTAB-GEN. The DTG decomposition curves of BNC-GEN and GEN showed peaks at 350 and 375 °C, while the slight increase in the peak of DTG compared to that in BNC (Figure 10b) indicates an interaction between BNC and genistein. For BNC-CTAB-GEN, the DTG curve showed two peaks at 254 and 337 °C, the former being related to the early degradation of alkyl chains of CTAB and carboxylic groups of TEMPO pretreatment, and the latter being related to unmodified parts of cellulose, as described before in BNC-CTAB (Figure 10b). Both curves indicate an interaction between compound and adsorbent, but despite this, the interaction between BNC-CTAB and genistein is stronger than that between BNC and genistein, due to the appearance of a shoulder on the DTG curve of BNC-GEN at 321 °C.

### 2.5. In Vitro Release Study in Gastrointestinal Fluids

The maximum desorption capacities for BNC-GEN and BNC-CTAB-GEN were 46.6843 and 66.4316 mg∙g^−1^, respectively. BNC-CTAB was found to release 1.42 times more genistein than BNC. Table 4 shows the adjustment of experimental data to kinetic models. For both systems, a good fit to the second linear order model was observed (R^2^ > 0.93) in the three simulated fluids. In stomach and small intestine fluids (pH 1.2 and 6.0, respectively), BNC-CTAB-GEN shows a higher desorption rate (h), while in colon fluid (pH 7.4), the BNC-GEN system shows higher h.

Moreover, the genistein release percentage from dried films of BNC-GEN and BNC-CTAB-GEN was determined and is shown in Figure 11. The results show that in the simulated stomach fluid, there is a sustained release of genistein that reaches almost 5% for both samples, while in small intestine fluid, approximately 20% of genistein is released in both samples; however, the release occurs in the first 5 min, and subsequently remains stable. Finally, the release reaches approximately 85% after 72 h in simulated colon fluid, but the release profile in colon fluid indicates a faster release in the BNC-GEN system and a controlled release for BNC-CTAB-GEN [44].

The low release rate in stomach fluid indicates that BNC and BNC-CTAB protect genistein from acidic conditions while genistein encapsulation protects its bioactivity during its transit through stomach fluids [14]. As both systems contain cellulose, they release a small amount of active compound in acid conditions; when the pH is 7, the release of genistein is favored by the swelling of the cellulose.

Previous results indicate that BNC and BNC-CTAB could act as protective agents of genistein in the stomach, aiding its subsequent release, where cells would be responsible for metabolizing genistein [10]; thus, BNC and BNC-CTAB can be considered for use as genistein nanocarriers.

## 3. Discussion

Colorectal cancer is one of the most malign and deadly carcinomas worldwide [3,7]. As a result, we are seeking to develop new strategies that inhibit the formation and growth of cancer [6]. Recently, the use of natural compounds has increased due to their selective and nontoxic characteristics [8]. Genistein is a promising chemopreventive agent of cancer, with different effects, including influence on inflammation, cell proliferation, angiogenesis inhibition, and antioxidant activity [27], related to the regulation of ER. Toxicity tests should be performed to determine the concentration at which active compounds affect cancer cells and their mechanisms of action. Previous studies by Wang et al. (2012) have demonstrated that genistein treatment induces G2 phase detention and the inhibition of cell proliferation in SW40 cells, because of an increase in DKK1 expression [45]. Additionally, the results presented by Sun et al. (2022) suggest that genistein induced SW620 cell cycle arrest in the G2/M phase by targeting mutant p53 [46]. In the study carried out by Quin et al. (2015), it was suggested that the downregulation of miR-95 and SGK1, and Akt phosphorylation, could be related to the antitumor effects of genistein in CRC [26]. According to the results obtained in this work, genistein presented similar IC50 values compared to those in the literature and greater selectivity to SW480 and SW620 CRC cell lines compared to healthy keratocytes. The results obtained in cellular assays determine the subsequent design of DDS; hence, genistein was encapsulated in BNC materials for its potential application in CRC chemoprevention.

BNC has been widely used in drug delivery systems to encapsulate active compounds [22] because its large surface area allows for the adsorption and further desorption of different compounds. However, surface modification is proposed in the literature to increase the adsorption of nonpolar compounds such as genistein, as the higher degree of sample hydrophobicity is related to an increase in the stability and biodisponibility of active compounds in aqueous systems [14,23]. The diameters found for BNC nanoribbons correspond to the typical morphology of BNC reported previously by Castro et al. (2012) [47]. For BNC after CTAB modification, the diameter increases slightly; this corresponds to the presence of CTAB molecules and was previously reported by K. Syverud et al. (2011) and N. Zainuddin et al. (2017) upon the surface modification of cellulose nanofibers and nanocrystals, respectively [37,48]. However, the structure of BNC was not altered, and a high surface area after modification was still available for the adsorption and subsequent release of genistein, which is ideal for the development of drug delivery systems. Additionally, the decrease in the hydrophilicity of the BNC-CTAB surface is attributed to the adsorption of CTAB on the surface of BNC-TEMPO by ionic bonding; the COO^−^ groups act as active sites for the adsorption of CTA^+^ polar heads [32].

Adsorption studies have suggested that BNC-CTAB possesses more binding sites that are available for genistein adsorption compared to BNC. However, isotherm models for BNC-CTAB showed a decrease in adsorption capacity when the temperature was increasing, and this could be attributed to the modified BNC properties; temperature reductions could increase sample wetting, and as a result, the increase in solute transfer from the liquid phase to the adsorbent surface occurs [49]. Moreover, thermodynamic parameters showed a slight decrease in terms of the Δ*G* values for BNC-CTAB-GEN compared to BNC-GEN, due to the energy of the hydrophobic interactions, which was lower than the hydrogen bond energy, and the increase in Δ*H* occurred because the BNC-CTAB-GEN system needs higher energy given that the intermolecular interactions are weaker.

The period of release assay was prolonged by 72 h because of its application in delivery systems for CRC, which are desired considering that the transit through the gastrointestinal tract could take time. The BNC and BNC-CTAB films do not show a burst release of genistein during transit through stomach and colon fluids. Additionally, the results show an increase in genistein desorption with increasing pH, until reaching a plateau after 48 h in the BNC-GEN system. In light of previous results, BNC and BNC-CTAB films acted as a carrier for genistein until it reached the intestinal fluids, where genistein is metabolized by cells [10]. Considering the IC50 values, 78.73 and 24.13 mg of BNC-GEN films, and 74.24 and 32.90 mg of BNC-CTAB-GEN are needed to reach a therapeutic concentration for SW480 at 24 and 48 h, respectively. For SW620, the amounts necessary are 76.91 and 43.71 mg of BNC-GEN films and 72.52 and 59.60 mg of BNC-CTAB-GEN films.

Finally, the modification of BNC with CTAB improves the adsorption profile of genistein, increases the loading capacity of the compound in the nanostructure from 53.91% to 71.73%, and provides stability during desorption, indicating a controlled and sustained release over time. Therefore, BNC surface modification could lead to favorable changes in the bioavailability of genistein, positively impacting the chemopreventive effect of the compound.

## 4. Materials and Methods

### 4.1. Materials

For the inoculum, glucose, peptone, yeast, sodium dihydrogen phosphate (NaH_2_PO_4_), potassium dihydrogen phosphate (KH_2_PO_4_), magnesium sulfate (MgSO_4_), and citric acid (*Komagataeibacter medellinensis* strain) were used, all of analytical grade. For the commercial medium of BNC, raw cane sugar and acetic acid in food grade were used. In addition, cetyl trimethyl ammonium bromide (CTAB) and ethanol 96% were used for the synthesis and modification of BNC; all of these were of analytical grade. Genistein from Shanghai Yingrui Biopharma Co (CAS 446-72-0) was used with a purity ≥ 98%. Finally, for the in vitro studies, the reagents used were sodium taurocholate, pepsin, sodium chloride (NaCl), maleic acid, sodium hydroxide (NaOH), potassium chloride (KCl), and disodium phosphate (Na_2_HPO_4_) of analytical grade, and soy lecithin of food grade. Trichloroacetic acid (TCA) at 50 wt.%, sulforhodamine B, glacial acetic acid, and Tris base were employed (of cell culture grade), and SW480 (CCL-228), SW620 (CCL-227), and HaCaT (PCS-200-011) lines were used.

### 4.2. Calculation of Inhibitory Concentration (IC50) for Free Genistein

Cell studies were used to prove the inhibitory effect of free genistein. The cells tested in the experiment were the following: SW480 (adenocarcinoma colorectal cancer cells), SW620 (metastatic colorectal cancer cells) and HaCaT (non-malignant human Keratinocytes). The experiments were conducted under the protocols of Agudelo et al. (2017) [50]. Briefly, cells were seeded in 96-well culture plates at a concentration of 20000 cells per well for cancer cells and 15000 for HaCaT and cultured at 37 °C in 5 vol.% CO_2_. After 24 h seeding, the cells were exposed to 7 different concentrations starting at 150 µM of genistein and incubated for 24 and 48 h. Cell monolayers were fixed to the well bottoms by adding 50 µL of 50 wt.% trichloroacetic acid (TCA) in each well, and the plates were incubated at room temperature for 1 h. The wells were then drained, rinsed twice with distilled water, and air dried. Sulforhodamine B (SRB) (0.4% w/v in 1 vol.% glacial acetic acid) was then added (100 µL/well), and the plates were incubated for 30 min. Unbound dye was drained and removed by washing 5 times with 1 vol.% glacial acetic acid. After air-drying the plate overnight, the dye was solubilized by adding 100 µL/well of 10 mM Tris base and stirred for 30 min at 37 °C. Absorbance at 490 nm was measured. All experiments were performed in quintuplicate. The absorbance of the control group (non-treated cells) was considered as 100% viability [50].

The percent inhibition was calculated using the following equation:(1)$$Inhibittion\left(\%\right)=\left[1-\frac{OD_{T}}{OD_{c}}\right]*100$$
where *OD_T_* is the optical density (OD) of treated cells and *OD_c_* is that for control (non-treated cells). The concentration able to inhibit 50% of cells (IC50) was calculated using the 4-parameter Hill equation and nonlinear regression. As proposed by Bertrand et al. (1992), Hill coefficients were found after plotting dose–response curves on the logarithmic scale [51]. Selectivity was calculated according to Equation (2).
(2)$$S=\frac{IC50_{Non-cancer\;cells}}{IC50_{Cancer\;cells}}$$

### 4.3. Development of BNC and BNC-CTAB Materials

#### 4.3.1. BNC Synthesis

BNC was synthesized from a non-commercial culture medium with raw cane sugar at 13 wt.% at pH 3.6, to which was added bacterial inoculum of *Komagataeibacter medellinensis* in plastic containers of 500 mL. This was incubated at room temperature for 15 days, and then, membranes were treated with 5 wt.% KOH for 14 h and washed to reach neutral pH. BNC was processed using a Super Masscolloider (MKCA 6-2) for the individualization of nanoribbons, and finally sterilized for subsequent use; the final concentration of the cellulose suspension was 1.37 wt.%.

#### 4.3.2. BNC Surface Modification

A preliminary modification of BNC was performed with TEMPO, as described by Cañas-Gutiérrez et al. (2020) [52]; 1 g of dry BNC was suspended in 500 mL of distilled water with 17 mg of TEMPO and 170 mg of NaBr to acquire TEMPO-oxidized BNC. To initiate the oxidation reaction, a NaClO solution (10 mmol∙g^−1^ cellulose) was added slowly to the TEMPO-oxidized BNC under stirring. The pH value was kept constant at 10.0 by adding NaOH 0.5M. Finally, the reaction was ceased by ethanol addition to the suspension. Products were washed with distilled water until neutral pH was reached. For the final modification, a CTAB solution (5mM) was prepared by diluting CTAB in deionized water. Later, 83 mL of the CTAB solution was added dropwise into 130 mL of TEMPO-oxidized BNC at 1.92% under magnetic stirring at room temperature. The mixture was heated to 60 °C for 30 min and then cooled down to room temperature. Modified cellulose (BNC-CTAB) was washed in dialysis membranes of 12–14 kD several times until unbound CTAB was removed and neutral pH reached. Therefore, no toxic leachate was generated.

#### 4.3.3. Morphological Analysis

The fibrillar structure and morphology of BNC and BNC-CTAB were analyzed by Transmission Electron Microscopy (TEM). A 10 µL drop was deposited on a Formvar/Carbon 200 Mesh Copper grid and stained with 2% uranyl acetate. The images were taken with a TECNAI FEI microscope operating at 80 kV at magnifications of 7.00 kX to 43.00 kX. Finally, the software ImageJ was used to measure the number of pixels and calculate the nanoribbons’ diameters; 60 measurements were taken per sample, and the results were compared using one-way analysis of variance (ANOVA). The statistical analysis was performed using RStudio.

#### 4.3.4. Contact Angle Measurements

To evaluate the surface hydrophobicity of the dry films of BNC and BNC-CTAB, contact angle measurements were made. For this, a flat surface of the material was placed on a goniometer coupled to a Dataphysics OCA 15EC camera. The system was calibrated using the ASTM D7490 08 standard; after that, a deionized 8 µL water drop was deposited, and with the help of the software, the contact angle was measured. The test was performed five times in different areas of the material.

#### 4.3.5. Chemical Analysis

To evaluate surface chemical changes, the infrared spectra of BNC and BNC-CTAB were obtained using a Nicolet 6700 spectrophotometer in ATR mode on a type IIA diamond crystal. The sample area was 0.5 mm^2^, and constant pressure was applied to each sample. Infrared spectra were collected between 4000 and 400 cm^−1^ with a resolution of 4 cm^−1^.

#### 4.3.6. Thermal Analysis

Thermal degradation of the samples was evaluated using a thermogravimetric analyzer (Mettler Toledo TGA/SDTA 851E). A total of 8 mg of the dried sample was weighed before and after the surface modification and heated in a nitrogen atmosphere from 30 to 800 °C, with a heating rate of 10 °C·min^−1^.

### 4.4. Adsorption Studies

#### 4.4.1. Genistein Quantification

Successive solutions were prepared from a genistein stock solution (25.00 mg∙L^−1^). Absorbance values were obtained by UV–Vis spectroscopy at a wavelength of 260 nm on a UV–Vis Evolution 600 spectrophotometer. A curve with 6 points was made, and using the linear regression function, slope and intercept values were found. The equation was y = 0.127x + 0.032 with a correlation coefficient of 0.999. These values allowed us to determine the concentration of genistein in a subsequent test.

#### 4.4.2. Adsorption Isotherms

A genistein stock solution (200 µg GEN∙mL^−1^ethanol) was initially prepared. Different mixtures of 10 mL were prepared as described in Table 5 and then placed in water baths at temperatures of 0, 23, and 40 °C.

After 2 h, samples were vacuum-filtered; 600 µL of this solution was diluted in 10 mL of ethanol to determine genistein concentration by UV–Vis spectroscopy. The experimental adsorption capacity *Q_t_* (mg∙g^−1^) of genistein was calculated by the following equation [53]:(3)$$Q_{t}=\frac{\left(C_{0}-C_{t}\right)V_{i}}{W}$$
where *C*_0_ is the initial concentration of genistein in the solution (mg∙L^−1^), *C_t_* is the genistein concentration at instant *t* (mg∙L^−1^), *V_i_* is the volume of solution during adsorption isotherms assay (L), and *W* is the weight of BNC and BNC-CTAB in solution (mg). If the adsorption process is long enough, *Q_t_* and *C_t_* will be constant and can be referred to as *Q_e_* and *C_e_*, corresponding to equilibrium adsorption capacity (mg∙g^−1^) and genistein concentration at equilibrium (mg∙L^−1^) [53]. Homogenous monolayer, heterogeneous monolayer, and multilayer (Appendix B) adsorption models were selected to analyze experimental data as described in Table 6.

#### 4.4.3. Determination of Thermodynamic Parameters

The thermodynamic parameters reflect the feasibility and spontaneity of the adsorption process, and can be evaluated through Gibbs free energy, enthalpy, and entropy, which can be calculated according to the method proposed by Pérez et al. (2011) [54]:(4)$$\Delta G{}^{\circ}=-RT\ln\left(K_{c}\right)$$
(5)ΔG°=ΔH°−TΔS°
where *R* (8.314 J∙mol^−1^∙K^−1^) is the universal gas constant, *T* (K) is the absolute temperature, ∆*G*° (kJ∙mol^−1^) is the change in Gibbs free energy, ∆*H* ° (kJ∙mol^−1^) is the enthalpy change, ∆*S*° (J∙mol^−1^∙K^−1^) is the entropy change, and $K_{c}$ is the equilibrium constant calculated using Equation (6) [53,54]:(6)$$K_{c}=\frac{C_{ad}}{C_{e}}$$

$C_{ad}$ is the concentration of the adsorbate contained on the surface of the adsorbent in equilibrium (mg∙L^−1^). For the Langmuir isotherm, the equilibrium constant $K_{c}$ is equal to $K_{L}$ (L∙mol^−1^) [55]. Arranging Equations (4) and (5), the Van’t Hoff equation is obtained [40,54]:(7)$$\ln\left(K_{c}\right)=\frac{-\Delta H{}^{\circ}}{RT}+\frac{\Delta S{}^{\circ}}{R}$$
plotting $\ln\left(K_{c}\right)$ vs. T^−1^, where the linear intercept is $\frac{\Delta S{}^{\circ}}{R}$ and the slope is $\frac{\Delta H{}^{\circ}}{R}$.

#### 4.4.4. Adsorption kinetics

Adsorption kinetics were determined by mixing BNC and BNC-CTAB at 0.5 wt.% with genistein at 42.28 µg∙mL^−1^ in a 45 mL conic tube. Tubes were placed in a water bath at 23 °C for 2 h, and 2 mL aliquots were taken at 5, 10, 20, 40, 60, 80, 100, 110, and 120 min. Samples were vacuum-filtered, and 600 µL of this solution was diluted in 10 mL of ethanol to determine genistein concentration in the solution via UV–Vis spectroscopy. The measurement was performed at a wavelength of 260 nm on a UV–Vis Evolution 600 spectrophotometer.

The experimental data were modeled using the following models:Pseudo-first-order—This model assumes that there is an adsorption site in the adsorbent for each adsorbate molecule. The kinetics are described by the following equation [53],
(8)$$Q_{t}=Q_{e}\left(1-e^{-K_{1}t}\right)\;$$
where *K*_1_ is the pseudo-first-order constant (min^−1^) and t is the time (min);

Pseudo-second-order—This model assumes that the adsorbate is adsorbed onto two active sites [56]. It is described by the following equations [53,56],

(9)$$\frac{t}{Q_{t}}=\frac{1}{K_{2}Q_{e}^{2}}+\frac{t}{Q_{e}}$$(10)$$Q_{t}=\frac{K_{2}Q_{e}^{2}t}{1+K_{2}Q_{e}t}\;$$
where *K*_2_ is the pseudo-second-order constant (g∙mg^−1^∙min^−1^). Equation (10) corresponds to the standard form of the model. Initial adsorption rate h (mg∙g^− 1^∙min^− 1^) can be determined as shown in the following equation [57],
(11)$$h=K_{2}Q_{e}^{2}$$

Elovich—This model assumes that the adsorbent active sites are heterogeneous; thus, it shows different activation energies [53,55,56]. It is described by the following equation [53,55,56],

(12)$$Q_{t}=\frac{1}{\beta }\ln\left(\alpha \beta \right)+\frac{1}{\beta }\ln\left(t\right)\;$$
where α (mg∙g^−1^∙min^−1^) and β (g∙mg^−1^) are the Elovich constants corresponding to the initial rate of adsorption, surface coverage, and activation energy;

Intra-particle diffusion—This model assumes the transfer of the adsorbate through the internal structure of the adsorbent; therefore, the adsorbent acquires a homogenous structure [55,56]. It is described by the following equation [55,56],

(13)$$Q_{t}=K_{3}\sqrt{t}$$
where $K_{3}$ is the diffusion constant (mg∙g^−1^∙min^−1/2^).

### 4.5. Development of Thin-Film Drug Delivery Systems

Thin films were prepared by adding 2.11 mL of GEN (42.28 mg∙L^−1^) into 7.89 mL of adsorbent solution (0.63 wt.%) at 23 °C. After 2 h, the samples were vacuum-filtered, and films were obtained. The drug delivery films were dried oven gravimetrically at 60 °C.

#### Thin Films’ Characterization

Atomic force microscopy (AFM) was used to characterize the surface morphology and texture of the films. Square thin films of 1 cm^2^ were analyzed using a Nanosurf FlexAFM mounted on an isostatic table. Images were recorded under static force of 10 nN, PID (100-800-100) in a 6.25 µm^2^ area using a triangle cantilever (HYDRA-ALL-G D) from AppNano. The recorded images were analyzed to determine the area roughness using C3000i software tools, and they were exported using the Gwyddion software. Five samples of BNC and BNC-CTAB were tested. Furthermore, films were analyzed by FTIR and TGA, as described in Section 4.3.5 and Section 4.3.6.

### 4.6. In Vitro Gastrointestinal Fluids Release Study

For genistein release profiles, the membrane dialysis method [58,59] was used in simulated fluids of the stomach, small intestine, and colon with pH values of 1.2, 6.0, and 7.4, respectively [14]. The dry thin films of BNC-GEN and BNC-CTAB-GEN were placed inside the dialysis membranes (MWCO: 12–14 kD). The sealed membranes were put into 78 mL of simulated fluid at 37 °C with a stirring speed of 80 rpm. During the first 120 min, the membranes remained in stomach fluid, from 120 to 1920 min in the small intestine fluid, and from 1920 to 4320 min in the colon fluid. The aliquots were taken, and immediately, the volume was replaced with the corresponding fluid. Genistein concentration was determined by UV–Vis spectroscopy at a wavelength of 260 nm by diluting 300 µL of aliquots in 5 mL of ethanol.

The release profile (%) of the adsorbate is described by the following equation [23]:(14)$$D=\frac{C_{d}}{C_{L}}*100$$
where *D* is the percentage of desorption, $C_{d}$ is the concentration of adsorbate in the release medium (mg∙L^−1^) and $C_{L}$ is the loaded concentration of the active compound in the adsorbent (mg∙L^−1^). The amount of solute released, $Q_{d}$, is described by:(15)$$Q_{d}=\frac{C_{d}V_{d}}{W}$$

The experimental data were modeled to pseudo-first- and pseudo-second-order kinetics according to Equations (8)–(10).

## 5. Conclusions

In this study, thin films of genistein loaded in BNC and BNC-CTAB were successfully prepared after the evaluation of surface modification, thermodynamic parameters, and adsorption. Several techniques, such as TEM, FTIR, TGA, and AFM, were used to analyze the modification of BNC and the correct adsorption of GEN on adsorbents. The results show an increase of two-fold in contact angle measurements as a result of BNC modification. The FTIR and TGA spectra reveal genistein incorporation into BNC and BNC-CTAB via the appearance of new peaks. The genistein successfully encapsulated in BNC and BNC-CTAB films was characterized by hydrogen bond interactions and electrostatic and hydrophobic interactions, respectively. Adsorption studies have demonstrated that the process was spontaneous and endothermic at all temperatures; however, the best adsorption capacity was reached at 23 °C. Furthermore, kinetics models indicate a better adjustment of BNC-CTAB, due to its hydrophobic character, which imparted genistein with more stability in aqueous systems. To conclude, in vitro studies showed a genistein release rate of 85% after 72 h and the low desorption of the compound into stomach fluid, thus indicating that BNC and BNC-CTAB acted as protective agents of genistein in acid conditions, and BNC and BNC-CTAB can be considered as genistein nanocarriers for CRC, as it can deliver concentrations with therapeutic effects, according to in vitro studies. Further work should be focused on animal evaluation and the pharmacokinetics of the system.

## Figures and Tables

**Figure 1 molecules-27-07201-f001:**
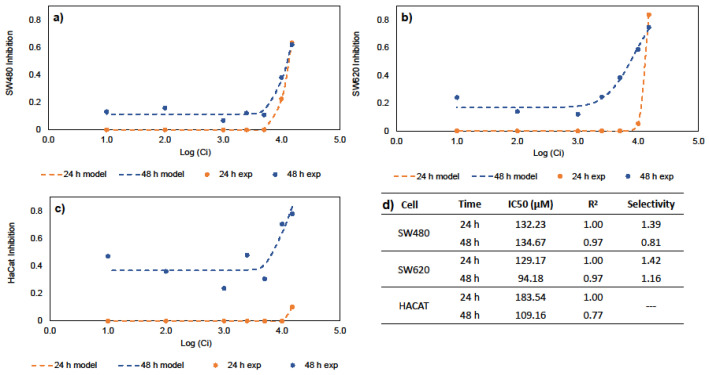
In vitro studies for IC50 and selectivity; (**a**) Hill equation for SW480 cells; (**b**) Hill equation for SW620 cells; (**c**) Hill equation for HaCaT cells; (**d**) IC50 for free genistein and selectivity.

**Figure 2 molecules-27-07201-f002:**
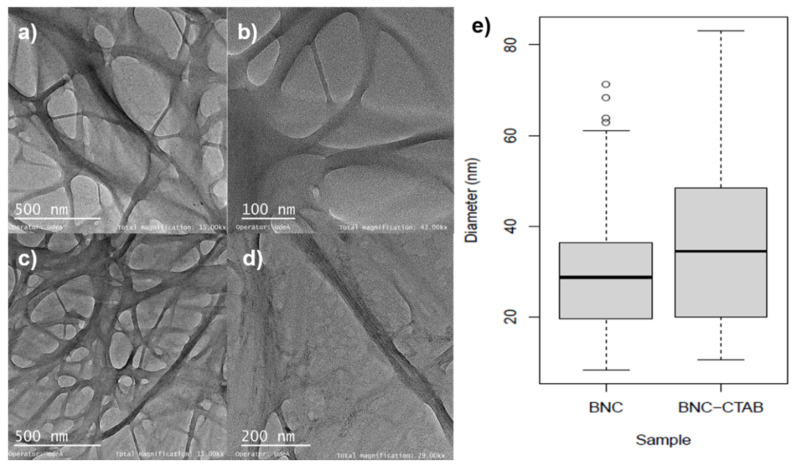
Morphological analysis (**a**) TEM micrographs of BNC at 15 kX; (**b**) TEM micrographs of BNC at 43 kX; (**c**) TEM micrographs of BNC-CTAB at 15 kX; (**d**) TEM micrographs of BNC-CTAB at 29 kX; (**e**) diameter distribution graphic.

**Figure 3 molecules-27-07201-f003:**
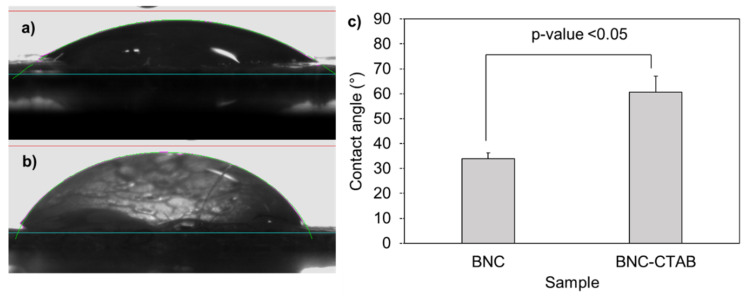
Contact angle experiments: (**a**) drop of deionized water in BNC; (**b**) drop of deionized water in BNC-CTAB; (**c**) average of contact angles. The sample groups were statistically different with *p* values < 0.05.

**Figure 4 molecules-27-07201-f004:**
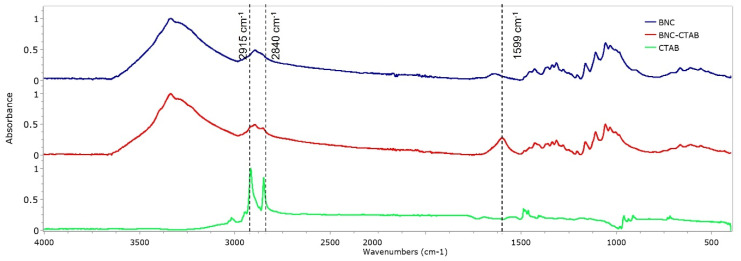
FTIR spectrum of BNC, BNC-CTAB, and free CTAB.

**Figure 5 molecules-27-07201-f005:**
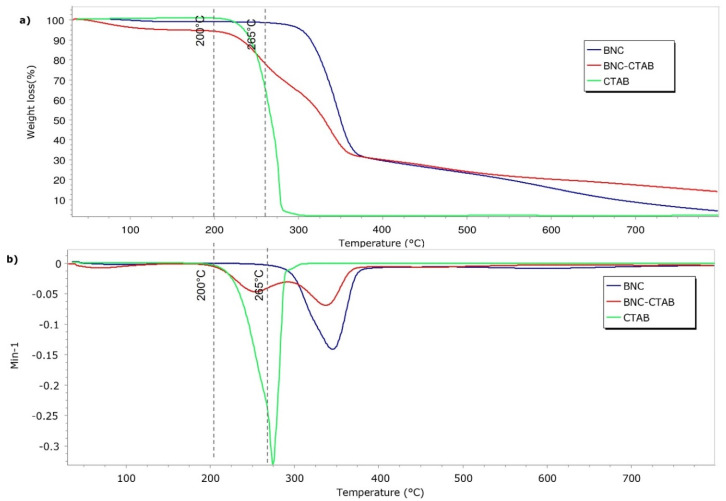
Thermogravimetric analysis of BNC, BNC-CTAB, and free CTAB: (**a**) TGA thermogram; (**b**) DTG curves.

**Figure 6 molecules-27-07201-f006:**
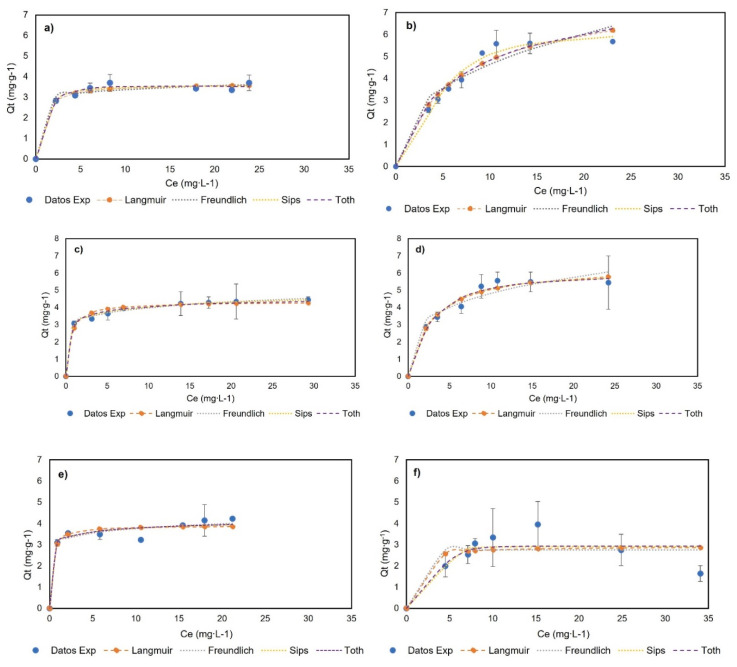
Adsorption isotherms of (**a**) BNC-GEN at 0 °C; (**b**) BNC-CTAB-GEN at 0 °C; (**c**) BNC-GEN at 23 °C; (**d**) BNC-CTAB-GEN at 23 °C; (**e**) BNC-GEN at 40 °C; (**f**) BNC-CTAB-GEN at 40 °C.

**Figure 7 molecules-27-07201-f007:**
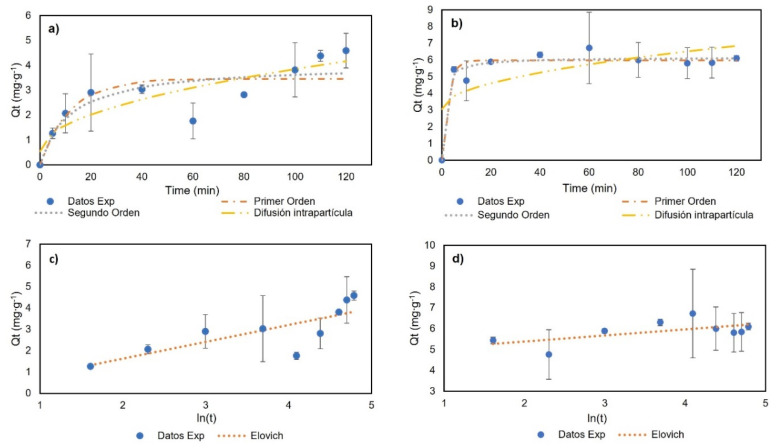
Adsorption kinetics results: (**a**) BNC-GEN; (**b**) BNC-CTAB-GEN; (**c**) BNC-GEN adjustment to Elovich model; (**d**) BNC-CTAB-GEN adjustment to Elovich model.

**Figure 8 molecules-27-07201-f008:**
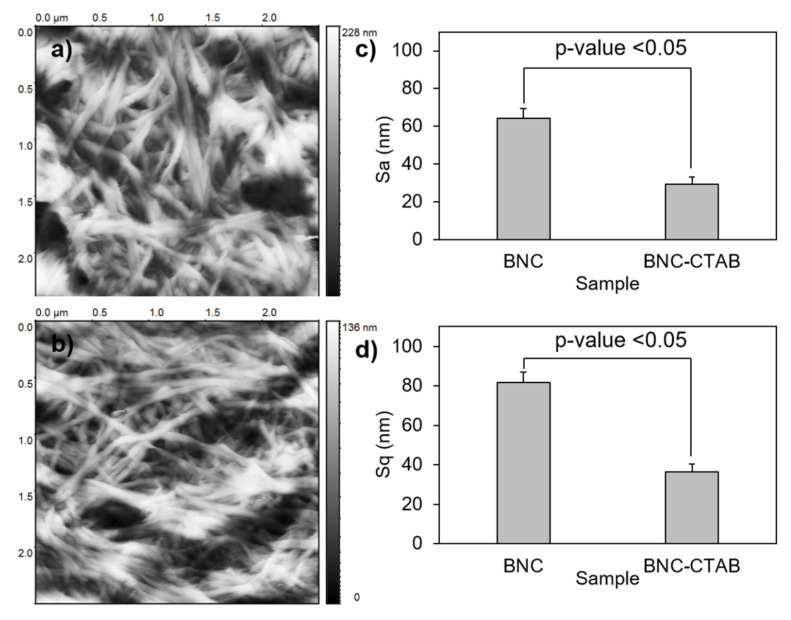
Roughness analysis; (**a**) AFM micrographs of BNC; (**b**) AFM micrographs of BNC-CTAB; (**c**) average of Sa; (**d**) average of Sq. The sample groups were statistically different with *p* values < 0.05.

**Figure 9 molecules-27-07201-f009:**
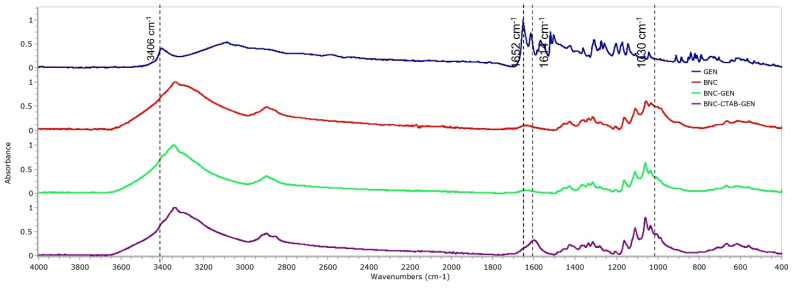
FTIR spectrum of GEN, BNC, BNC-GEN, and BNC-CTAB-GEN.

**Figure 10 molecules-27-07201-f010:**
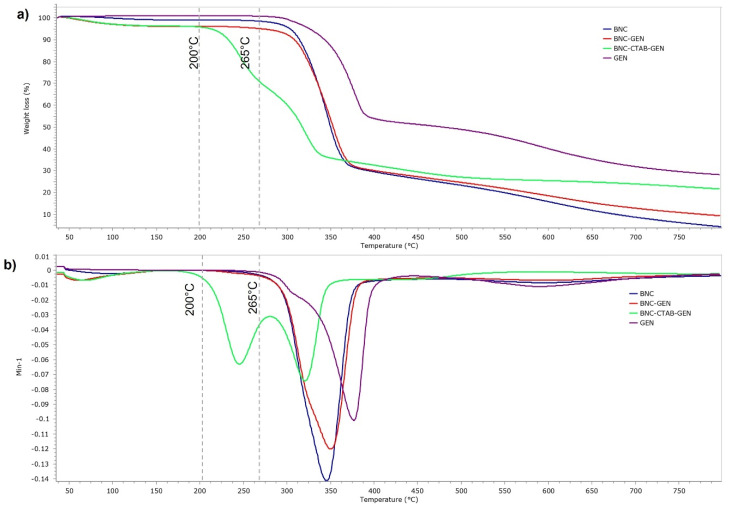
Thermogravimetric analysis of BNC, GEN, BNC-GEN, and BNC-CTAB-GEN; (**a**) TGA thermogram; (**b**) DTG curves.

**Figure 11 molecules-27-07201-f011:**
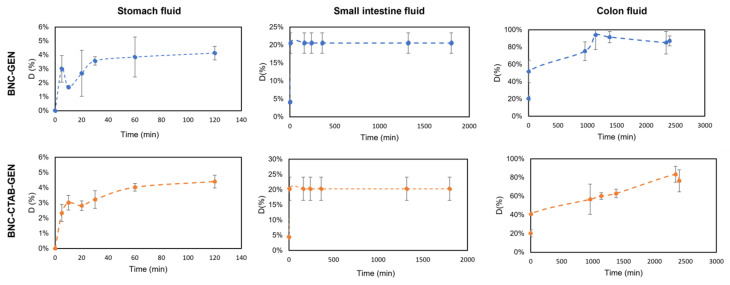
The release profile of genistein in gastrointestinal fluids.

**Table 1 molecules-27-07201-t001:** Isotherms models for genistein adsorption in BNC and BNC-CTAB.

Isotherm Models		BNC-GEN			BNC-CTAB-GEN		
		0 °C	23 °C	40 °C	0 °C	23 °C	40 °C
Langmuir	*Q_m_* (mg·g^−1^)	3.67	4.34	3.90	8.12	6.47	2.92
	*K_L_* (L·mg^−1^)	1.58	1.80	3.95	0.15	0.35	1.68
	R^2^	0.98	0.98	0.95	0.96	0.98	0.65
Freundlich	*K_F_*	8.12	3.04	3.11	1.88	2.63	2.75
	$n_{f}$	0.15	8.44	12.13	2.53	3.81	4.30 × 10^7^
	R^2^	0.96	0.99	0.97	0.92	0.95	0.64
Sips	*Q_mS_* (mg·g^−1^)	3.56	5.56	4.63	6.24	6.12	2.91
	*a_S_* (L·mg^−1^)	0.96	1.12	2.07	0.05	0.31	0.00
	*B_S_*	1.65	0.37	0.33	2.00	1.20	5.87
	R^2^	0.98	0.99	0.96	0.98	0.98	0.71
Toth	*Q_mT_* (mg·g^−1^)	4.47	4.84	4.99	6.55	5.99	2.93
	*A_T_* (L·mg^−1^)	0.65	3.48	8.49	0.24	0.17	0.00
	*z*	1.09	0.50	0.24	0.92	1.38	5.14
	R^2^	0.99	0.99	0.96	0.95	0.98	0.71

**Table 2 molecules-27-07201-t002:** Thermodynamic results for BNC-GEN and BNC-CTAB-GEN.

**BNC-GEN**			
**T (°C)**	**Δ*G* (kJ∙mol^−1^)**	**Δ*H* (kJ∙mol^−1^)**	**Δ*S* (J∙mol^−1^∙K^−1^)**
0	−29.45	15.21	162.58
23	−32.25		
40	−36.14		
**BNC-CTAB-GEN**			
**T (°C)**	**Δ*G* (kJ∙mol^−1^)**	**Δ*H* (kJ∙mol^−1^)**	**Δ*S* (J∙mol^−1^∙K^−1^)**
0	−24.099	41.36	238.33
23	−28.225		
40	−33.913		

**Table 3 molecules-27-07201-t003:** Adsorption kinetics models.

**Pseudo-First-Order**			**Pseudo-Second-Order**		
**Parameters**	**BNC-GEN**	**BNC-CTAB-GEN**	**Parameters**	**BNC-GEN**	**BNC-CTAB-GEN**
*Q_e_* (mg∙g^−1^)	3.46	5.98	*Q_e_* (mg∙g^−1^)	4.05	6.16
*K*_1_ (min^−1^)	0.08	0.39	*K*_2_ (g∙mg^−1^∙min^−1^)	0.02	0.15
R^2^	0.70	0.94	R^2^	0.75	0.96
**Elovich**			**Intra-Particle Diffusion**		
**Parameters**	**BNC-GEN**	**BNC-CTAB-GEN**	**Parameters**	**BNC-GEN**	**BNC-CTAB-GEN**
*α* (mg∙g^−1^∙min^−1^)	0.845	4.15 × 10^6^	*C* (mg∙L^−1^)	0.53	3.07
*β* (g∙mg^−1^)	1.27	3.431	*K*_3_ (mg∙g^−1^∙min^−1/2^)	0.33	0.35
R^2^	0.62	0.364	R^2^	0.78	0.48

**Table 4 molecules-27-07201-t004:** Release kinetics models.

Models		Stomach Fluid		Small Intestine Fluid		Colon Fluid	
		BNC-GEN	BNC-CTAB-GEN	BNC-GEN	BNC-CTAB-GEN	BNC-GEN	BNC-CTAB-GEN
PFO	*Q_d_* (mg∙g^−1^)	1.62	2.63	10.68	16.05	42.10	50.46
	*K*_1_ (min^−1^)	3.29	12.00	1.78	1.78	0.16	12.00
	R^2^	0.68	0.75	0.93	0.91	0.98	0.94
PSO	*Q_d_* (mg∙g^−1^)	2.19	3.64	11.68	17.32	44.46	63.70
	*K*_2_ (g∙mg^−1^∙min^−1^)	6.59 × 10^2^	4.11 × 10^−2^	6.18 × 10^−4^	4.88 × 10^−4^	4.28 × 10^−4^	9.38 × 10^−5^
	*h* (mg∙g^−1^∙min^−1^)	0.32	0.54	0.08	0.15	0.85	0.38
	R^2^	0.98	0.99	0.93	0.95	0.99	0.96

**Table 5 molecules-27-07201-t005:** Concentrations of genistein for adsorption isotherms assays.

Sample	Genistein Concentration (mg∙L^−1^)	Adsorbent Concentration (%)
GE_9_	65.34	0.5
GE_8_	51.48	0.5
GE_7_	42.28	0.5
GE_6_	38.64	0.5
GE_5_	35.00	0.5
GE_4_	26.72	0.5
GE_3_	23.26	0.5
GE_2_	19.80	0.5
GE_1_	16.34	0.5
GE_0_	0.00	0.5

**Table 6 molecules-27-07201-t006:** Adsorption models of homogeneous and heterogenous monolayers [38,53].

Model	Parameters	Description
Langmuir $Q_{e}=\frac{Q_{mL}K_{L}C_{e}}{1\;+\;K_{L}C_{e}}$	$Q_{mL}$, maximum adsorption capacity (mg∙g^−1^) $K_{L}$, Langmuir constant (L∙mg^−1^).	Assumes monolayer adsorption and homogeneous surface. Additionally, adsorption is localized.
Freundlich $Q_{e}=K_{f}C_{e}^{1/n_{f}}$	$K_{f}$, Freundlich constant (mg^1−n^∙L^n^∙g^−1^) $n_{f}$, adsorption intensity	Describes a heterogeneous surface of adsorption with the interaction between adsorbed molecules.
Sips $Q_{e}=\frac{Q_{ms}a_{s}C_{e}^{B_{s}}}{1\;+\;a_{s}C_{e}^{B_{s}}}$	$Q_{ms}$, maximum adsorption capacity (mg∙g^−1^) $a_{s}$, Sips constant (L∙mg^−1^) $B_{s}$, model exponent	Describes heterogeneous systems, localized adsorption without adsorbate–adsorbate interactions. $B_{s}$ is known as the heterogeneity factor.
Toth $Q_{e}=\frac{Q_{mT}C_{e}}{{\left(A_{T}\;+\;C_{e}^{z}\right)}^{\frac{1}{z}}}$	$Q_{mT}$, maximum adsorption capacity (mg∙g^−1^) $A_{T}$, Toth constant (L∙mg^−1^) z, model exponent	Describes heterogeneous systems. Parameter *z* is related to system heterogeneity.

## Data Availability

The data presented in this study are available upon request to the corresponding author.

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
