# Peer review of "Development of Genistein Drug Delivery Systems Based on Bacterial Nanocellulose for Potential Colorectal Cancer Chemoprevention: Effect of Nanocellulose Surface Modification on Genistein Adsorption"

_molecules, 2022, doi:10.3390/molecules27217201_

Round 1

Reviewer 1 Report

The manuscript deals with exploring the potential of Genestein DDS based on surface modified bacterial nanocellulose for colororectal cancer chemoprevention. The topic is interesting and the methodology is well described. I recommend publishing after minor revisions 

1. Abstract is too lengthy. Please focus on the work protocol and findings

2. Please elaborate more on the relation between surface modification and the colorectal cancer chemoprevention. Add more interpretation about the possible mechanism in the discussion part.

Author Response

We thank you for the careful reading of our manuscript entitled “Development of genistein drug delivery systems based on bacterial cellulose for potential colorectal cancer chemoprevention: effect of nanocellulose surface modification on genistein adsorption”, We thank reviewer 1 for all your comments, which help us to improve the quality of work. We present the replies to reviewer comments below:

Reviewer 1

Comment

Response

1. Abstract is too lengthy. Please focus on the work protocol findings.

Modified as requested

2. Please elaborate more on the relation between surface modification and the colorectal cancer chemoprevention. Add more interpretation about the possible mechanism in the discussion part.

Genistein is the compound that has anticancer and antioxidant properties, however, due to its solubility limitations, the use of CTAB was studied. Surface modification of bacterial cellulose was performed to increase the solubility, bioavailability, and stability of genistein in aqueous systems.

More interpretation was added in the discussion and can be seen in line 370.

In the document all changes were highlighted in red and the Track Changes is on.

Reviewer 2 Report

The article presents a study concerning the effect of a modification of the nanocellulose surface on the adsorption of Genistein, a natural molecule with anticancer activities. The authors wish to develop a delivery system of genistein based on nanocellulose as a carrier for colorectal cancer prevention. As mentioned by the authors, their preparation method was inspired by a strategy developed by Wang et al. 2019. The studied modification by the authors is based on the adsorption of CTAB on bacterial nanocellulose oxidized by TEMPO. The modification was evaluated by water drop contact angle measurement on dry nanocellulose films and by FTIR.  The adsorption of genistein on this CTAB-modified nanocellulose is determined by different techniques: adsorption studies to determine adsorption isotherms, thermodynamic parameters and adsorption kinetics, AFM analyses to characterize the roughness and FTIR analyses to verify the adsorption of genistein. Finally, the authors present results of in vitro release of genistein in fluids mimicking gastrointestinal conditions. The originality of the work lies in the theoretical study of the affinity of genistein with nanocellulose and the use of models.

There is a real need to develop solutions to prevent this type of cancer. Thus, it is essential to have a better understanding of the interactions between genistein and non-toxic bio-based carriers such as nanocelluloses. The manuscript has been carefully written, and the results are quite well commented. The subject is interesting.

The authors must check the bibliographic references.

Comments:

Line 36: the reference 1 is no longer available

Line 39: reference 5 no longer exists

Line 95: in paragraph 2.1, the authors are invited to explain a little more the principle of the evaluation of the inhibition of the growth of cancer cells

Line 108: redo the table to gain visibility and understanding: alignment of the lines in table 1.d

Line 119: correct media by median

Line 129:  it is more a question of a decrease of the hydrophilicity rather than a hydrophobization knowing that the hydrophobic character is commonly considered as soon as the drop angle becomes higher than 90°.

Line 159: add hashed lines corresponding to the respective degradation temperatures of BNC and BNC-CTAB and CTAB

Line 166 : the experimental data did not fit the multilayer models and can be seen in Appendix A

Line 167 : there is no Appendix A correct line 604

Line 169 : Is it really necessary to have three digits after the comma for concentrations?

Line 175: at 40°C, why do we observe an unstable adsorption? What happens with BNC-CTAB and Genistein at this temperature?

Line 188: standardize the writing of the abbreviation nf

Line 195: the numbers in column 7 are not perfectly centered

Line 229: replace Tiempo by time

Line 233: add majuscule to “the”

Line 262: correct the sentence: the spectrum of the BNC is not present in figure 8.

Line 268: the absence of the BNC-CTAB and BNC-CTAB-GEN spectra on the same figure prevents readers from really seeing the differences between the two spectra. The authors indicate that the differences are more in the order of the height and the ratio of the bands, but it is very difficult to realize this. The readers have to go back and forth between figure 4 and figure 9. Do the authors have the possibility to perform additional characterizations: by XPS to have the surface chemical composition of the materials or by measuring the surface zeta potential to confirm the presence of genistein?

Line 283: On the same note, it is very complicated to see the differences between BNC and BNC-GEN or BNC-CTAB and BNC-CTAB-GEN because the curves are not on the same figure. The authors talk about a shoulder that appears on the DTG curve of the BNC-GEN at 321°C, but it seems that this shoulder is also present on the BNC curve of figure 4. It would be nice to put the curves on the same figure

Line 304: It is stated in the text that the low release in gastric fluid indicates that BNC and BNC-CTAB protect genistein from acidic conditions. Can the authors elaborate further on this point?

Line 344: The toxicity of CTAB is not discussed in the text. It would have been interesting to test the cytotoxicity of BNC-CTAB. Similarly, the release of CTAB or BNC particles into gastrointestinal fluids is not discussed. Can you comment on these points?

Line 388: reference 50 does not match Agudelo et al 2017

Line 403: replace inhibittion by inhibition

Line 406: Hill's equation is not used to calculate the IC50 concentration but rather to estimate the number of ligand molecules that must bind to a receptor to produce a functional effect. The reference given "Elsevier Oceanogr. Ser. vol. 7, pp. 135-149, 1974, doi: 10.1016/S0422- 7359894(08)70981-4 is not consistent with the title of the reference. This is the reference Weiss JN. The Hill equation revisited: uses and misuses. FASEB Journal : Official Publication of the Federation of American Societies for Experimental Biology. 1997 Sep;11(11):835-841. DOI: 10.1096/fasebj.11.11.9285481. PMID: 9285481. But this reference does not explain the calculation of the IC50 concentration. Check the references and explain the method for calculating the IC50.

Line 564: replace the sentence: “The release profile (%) of and adsorbate is described by the following equation” by “The release profile (%) of the adsorbate is described by the following equation”

Line 604: Replace Appendix B by Appendix A

Line 729: the reference 50 and 51 are identical!

Line 735: the reference is not right.

Author Response

We thank you for the careful reading of our manuscript entitled “Development of genistein drug delivery systems based on bacterial cellulose for potential colorectal cancer chemoprevention: effect of nanocellulose surface modification on genistein adsorption”, We thank reviewer 2 for all your comments, which help us to improve the quality of work. We present the replies to reviewer comments below:

Reviewer 2

Comment

Response

1. the reference 1 is no longer available

Reference was check and is available.

2. reference 5 no longer exists

Reference was check and is available.

3. in paragraph 2.1, the authors are invited to explain a little more the principle of the evaluation of the inhibition of the growth of cancer cells.

Modified as requested

4. redo the table to gain visibility and understanding: alignment of the line in table 1.d

Modified as requested

5. correct media by median

Done as requested

6. it is more a question of a decrease of the hydrophilicity rather than a hydrophobization knowing that the hydrophobic character is commonly considered as soon as the drop angle becomes higher than 90°.

The term was corrected as requested

7. add hashed lines corresponding to the respective degradation temperatures of BNC and BNC-CTAB and CTAB

Figure 5 was modified and hashed lines were added

8. Is it really necessary to have three digits after the comma for concentrations?

Concentrations in paragraph 2.3.1 have been corrected to have two digits after the comma

9. at 40°C, why do we observe an unstable adsorption? What happens with BNC-CTAB and Genistein at this temperature?

In the adsorption isotherms at 40°C for BNC-CTAB experimental data did not present a sustained plateau. Also, deviations were higher compared to the others isotherms.

10. Line 188 standardize the writing of the abbreviation nf

Done as requested

11. Line 195 the numbers in column 7 are not perfectly centered

Done as requested

12. Line 229 replace Tiempo by time

Modified as requested

13. Line 233: add majuscule to the

Done as requested

14. Line 262: correct the sentence: the spectrum of the BNC is not present in figure 8.

BNC spectrum correspond to figure 9. The figure was modified

15. the absence of the BNC-CTAB and BNC-CTAB-GEN spectra on the same figure prevents readers from really seeing the differences between the two spectra. The authors indicate that the differences are more in the order of the height and the ratio of the bands, but it is very difficult to realize this. The readers have to go back and forth between figure 4 and figure 9. Do the authors have the possibility to perform additional characterizations: by XPS to have the surface chemical composition of the materials or by measuring the surface zeta potential to confirm the presence of genistein?

A modification in figure 9 was made, the BNC spectrum was added.

16. it is very complicated to see the differences between BNC and BNC-GEN or BNC-CTAB and BNC-CTAB-GEN because the curves are not on the same figure. The authors talk about a shoulder that appears on the DTG curve of the BNC-GEN at 321°C, but it seems that this shoulder is also present on the BNC curve of figure 4. It would be nice to put the curves on the same figure.

A modification in figure 10 was made, BNC curves were added to the figure.

9. It is stated in the text that the low release in gastric fluid indicates that BNC and BNC-CTAB protect genistein from acidic conditions. Can the authors elaborate further on this point?

Modified as requested. Explanation was added in line 305.

10. The toxicity of CTAB is not discussed in the text. It would have been interesting to test the cytotoxicity of BNC-CTAB. Similarly, the release of CTAB or BNC particles into gastrointestinal fluids is not discussed. Can you comment on these points?

After the surface modification modified cellulose were washed several times until CTAB excess was removed from solution, like that, no toxic leachate is formed.

Additionally, paragraph 4.3.2 was modified to clarify the comment.

11. reference 50 does not match Agudelo et al 2017

Reference was corrected

Line 403: replace inhibittion by inhibition

Done as requested

Line 406: Hill's equation is not used to calculate the IC50 concentration but rather to estimate the number of ligand molecules that must bind to a receptor to produce a functional effect. The reference given "Elsevier Oceanogr. Ser. vol. 7, pp. 135-149, 1974, doi: 10.1016/S0422- 7359894(08)70981-4 is not consistent with the title of the reference. This is the reference Weiss JN. The Hill equation revisited: uses and misuses. FASEB Journal : Official Publication of the Federation of American Societies for Experimental Biology. 1997 Sep;11(11):835-841. DOI: 10.1096/fasebj.11.11.9285481. PMID: 9285481. But this reference does not explain the calculation of the IC50 concentration. Check the references and explain the method for calculating the IC50

Thank you for the comment. A new reference was added and line 411 were modified as requested.

Line 564: replace the sentence: “The release profile (%) of and adsorbate is described by the following equation” by “The release profile (%) of the adsorbate is described by the following equation”

Replaced as requested

Line 604: Replace Appendix B by Appendix A

Modified as requested

Line 729: the reference 50 and 51 are identical!

Both references were corrected

Line 735: the reference is not right.

Reference was verified and corrected as correspond

In the document all changes were highlighted in red and the Track Changes is on.